# Vaccination with mycobacterial lipid loaded nanoparticle leads to lipid antigen persistence and memory differentiation of antigen-specific T cells

Eva Morgun[1], Jennifer Zhu[2], Sultan Almunif[2], Sharan Bobbala[2†], Melissa S Aguilar[3], Junzhong Wang[1‡], Kathleen Conner[1], Yongyong Cui[1], Liang Cao[1], Chetan Seshadri[3], Evan A Scott[1]*, Chyung-Ru Wang[1]*

[1]Department of Microbiology and Immunology, Feinberg School of Medicine, Northwestern University, Chicago, United States; [2]Department of Biomedical Engineering, Northwestern University, Evanston, United States; [3]Department of Medicine, University of Washington School of Medicine, Seattle, United States

*For correspondence:
evan.scott@northwestern.edu
(EAS);
chyung-ru-wang@northwestern.
edu (C-RW)

Present address: †Department of
Pharmaceutical Sciences, School
of Pharmacy, West Virginia
University, Morgantown, United
States; ‡Department of Infectious
Disease, Union Hospital of Tongji
Medical College, Huazhong
University of Science and
Technology, Wuhan, China

Competing interest: See page
17

Reviewing Editor: Mone Zaidi,
Icahn School of Medicine at
Mount Sinai, United States

**Abstract** *Mycobacterium tuberculosis* (Mtb) infection elicits both protein and lipid antigen-specific T cell responses. However, the incorporation of lipid antigens into subunit vaccine strategies and formulations has been underexplored, and the characteristics of vaccine-induced Mtb lipid-specific memory T cells have remained elusive. Mycolic acid (MA), a major lipid component of the Mtb cell wall, is presented by human CD1b molecules to unconventional T cell subsets. These MA-specific CD1b-restricted T cells have been detected in the blood and disease sites of Mtb-infected individuals, suggesting that MA is a promising lipid antigen for incorporation into multicomponent subunit vaccines. In this study, we utilized the enhanced stability of bicontinuous nanospheres (BCN) to efficiently encapsulate MA for in vivo delivery to MA-specific T cells, both alone and in combination with an immunodominant Mtb protein antigen (Ag85B). Pulmonary administration of MA-loaded BCN (MA-BCN) elicited MA-specific T cell responses in humanized CD1 transgenic mice. Simultaneous delivery of MA and Ag85B within BCN activated both MA- and Ag85B-specific T cells. Notably, pulmonary vaccination with MA-Ag85B-BCN resulted in the persistence of MA, but not Ag85B, within alveolar macrophages in the lung. Vaccination of MA-BCN through intravenous or subcutaneous route, or with attenuated Mtb likewise reproduced MA persistence. Moreover, MA-specific T cells in MA-BCN-vaccinated mice differentiated into a T follicular helper-like phenotype. Overall, the BCN platform allows for the dual encapsulation and in vivo activation of lipid and protein antigen-specific T cells and leads to persistent lipid depots that could offer long-lasting immune responses.

## eLife assessment

The authors generate a new formulation built upon a previous nanoparticle platform to generate a new system termed bicontinuous nanospheres (BCN), allowing for the dual incorporation of lipid and protein antigens. The authors generate mycolic acid (MA)-loaded BCN and perform a series of characterization studies to demonstrate the superior performance of this new formulation relative to the original one in terms of antigen persistence, a quality needed to sustain responses after vaccination. This work provides **important** new insights relevant to the TB vaccine field and it suggests that alternative antigens to proteins could be used in TB vaccine formulations. The data are **convincing** and will be of interest to individuals working on tuberculosis, vaccines and basic immunology.

## Introduction

Vaccination is currently the most effective method for the prevention and eradication of infectious diseases. In particular, the development of subunit vaccine formulations that include only immuno-dominant antigens from pathogens paired with select adjuvants have contributed to these efforts. In contrast to attenuated and inactivated vaccines that contain diverse molecular components of pathogens, subunit vaccines are often preferred due to their simplicity, safety, stability, and scalability of production (*Karch and Burkhard, 2016*). Yet the development of effective vaccines for several key endemic parasitic and bacterial pathogens has remained elusive. Antigen selection for subunit vaccines has thus far focused on protein molecules, overlooking a vast family of non-protein antigens such as lipids, glycolipids, metabolites, and phosphoantigens, which can be found on and within pathogens and can be recognized by various subsets of unconventional T cells (*Legoux et al., 2017*).

Persistent antigen exposure is known to occur naturally after infection through antigen archiving by follicular dendritic cells (FDCs), which periodically present antigens to cognate B cells and aid in affinity maturation and somatic hypermutation (*Heesters et al., 2014*; *Hirosue and Dubrot, 2015*). Antigen persistence can also support the maintenance of T cell response through the periodic priming of circulating T cells (*Takamura et al., 2010*; *Kim et al., 2010*; *Demento et al., 2012*; *Zammit et al., 2006*; *Woodland and Kohlmeier, 2009*; *Vokali et al., 2020*). Subcutaneous vaccination has also been shown to lead to the archiving of protein antigens within lymphatic endothelial cells (LECs), which induce the development of protective T cell immunity (*Tamburini et al., 2014*; *Walsh et al., 2021*). However, little is known about how other vaccination routes and vaccine formulations may affect antigen archiving and persistence.

Lipid and glycolipid antigens can be presented by group 1 (CD1a, b, and c) and group 2 (CD1d) CD1 molecules to CD1-restricted T cells. Unlike MHC molecules, CD1 molecules exhibit limited polymorphism (*Reinink and Van Rhijn, 2016*). Therefore, CD1-restricted microbial lipid antigens are likely to be recognized by most individuals, making them attractive targets for vaccine development (*Joosten et al., 2019*). Group 1 CD1-restricted T cells have diverse T cell receptors and can recognize a variety of lipid and glycolipid antigens (*Morgun et al., 2021*). Microbial lipid-specific group 1 CD1-restricted T cells have been identified from several pathogens, including Mtb and other mycobacterial species, *Staphylococcus aureus* (SA), *Borrelia burgdorferi*, *Salmonella typhimurium*, and *Brucella melitensis* (*Morgun et al., 2021*). CD1b-restricted T cells specific to Mtb lipids including mycolic acid (MA), an immunodominant lipid antigen (*Busch et al., 2016*) and the component of the mycobacterial cell wall, have been identified in patients with tuberculosis (TB) (*Lopez et al., 2020*; *Montamat-Sicotte et al., 2011*). As group 1 CD1 molecules are not expressed in mice, we have previously developed a human CD1 transgenic mouse model (hCD1Tg). (*Felio et al., 2009*) Using hCD1Tg mice, we have shown that transgenic CD1b-restricted DN1 T cells specific for MA could protect against Mtb infection (*Zhao et al., 2015*), making these potential antigens of interest in a subunit vaccine against TB.

The primary goal of a vaccine is to induce the differentiation of the adaptive immune system such that upon re-exposure to the antigen, a faster and larger response occurs, halting pathogen spread. While conventional T cells and B cells are known to undergo this differentiation, unconventional T cells such as NKT cells do not possess a comparable memory phenotype. The presence of Mtb lipid-specific group 1 CD1-restricted memory T cells has been described in both human and mice (*Lopez et al., 2020*; *Montamat-Sicotte et al., 2011*; *Felio et al., 2009*). However, the properties of memory group 1 CD1-restricted T cells, which share a similar developmental pathway as NKT cells, are not known.

Lipid antigens are difficult to formulate and administer due to their hydrophobic nature. To assess the utility of MA as a component of a subunit vaccine, we previously encapsulated MA in poly(ethylene glycol)-block-poly(propylene sulfide) (PEG-*b*-PPS) micellar nanocarriers (MA-MC) (*Shang et al., 2018*). Nanocarriers composed of PEG-*b*-PPS possess several unique benefits compared to other delivery platforms, including high stability in aqueous solution (*Bobbala et al., 2020*), minimal background immunostimulation (*Burke et al., 2022*), and triggered release of cargo when exposed to physiological levels of reactive oxygen species (*Du et al., 2017*; *Scott et al., 2012*; *Napoli et al., 2004*). We showed that MA-MC could effectively activate adoptively transferred DN1 T cells and elicit polyclonal group 1 CD1-restricted T cell response in hCD1Tg mice after intranasal delivery (*Shang et al., 2018*). While effective for stable loading of lipophilic and amphiphilic payloads, micellar nanocarriers, do not allow for facile encapsulation of hydrophilic payloads like protein antigens.

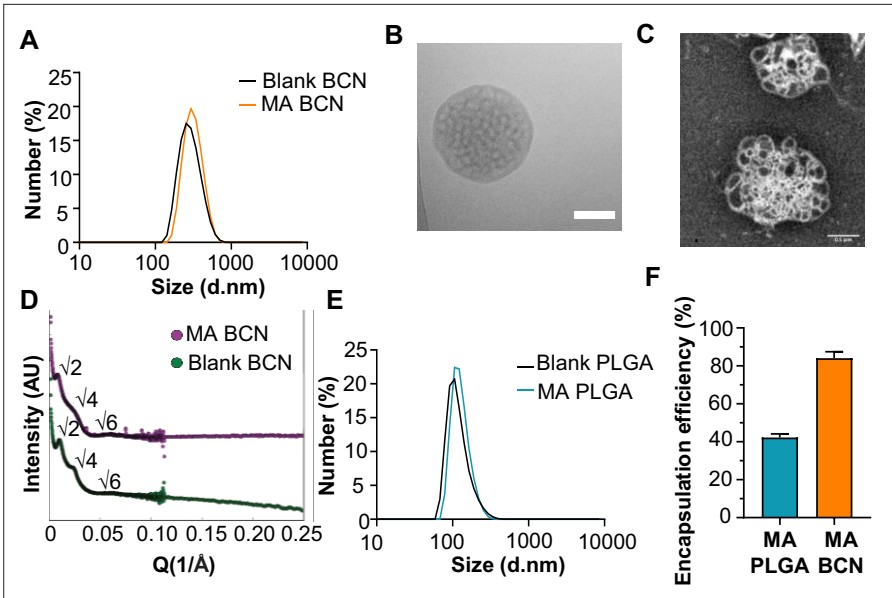

**Figure 1.** Physicochemical characterization of bicontinuous nanospheres (BCN) and PLGA nanocarrier formulation (PLGA-NP). (**A**) Dynamic light scattering (DLS) analysis of blank BCN and MA-loaded BCN (MA-BCN). (**B**) Cryo-TEM of MA-BCN (scale = 500 nm). (**C**) Negative staining transmission electron microscopy (TEM) images of MA-BCN. (**D**) Small angle X-ray scattering (SAXS) of blank BCN and MA-BCN. (**E**) Dynamic light scattering (DLS) analysis of blank poly(D,L-lactide-co-glycolide) (PLGA) and MA loaded poly(D,L-lactide-co-glycolide) (MA PLGA). (**F**) MA encapsulation efficiency for MA-BCN and MA PLGA. N=3 per condition. Data represented as mean ± SD.

The online version of this article includes the following figure supplement(s) for figure 1:

**Figure supplement 1.** Physicochemical characterization of PLGA nanocarrier formulation (PLGA-NP).

**Figure supplement 2.** Intracellular localization of poly(D,L-lactide-co-glycolide) nanoparticle (PLGA-NP) and bicontinuous nanospheres (BCN).

Here, we build upon our prior lipid antigen vaccine formulation by employing a nanocarrier platform engineered for the dual loading of both lipid and protein antigens. BCN are polymeric analogs to lipid cubosomes and thus possess a highly organized internal cubic organization containing both intertwined hydrophobic bilayers for lipid loading as well as hydrophilic aqueous channels for protein loading (*Allen et al., 2019*). Using the flash nanoprecipitation (FNP) technique (*Saad and Prud'homme, 2016*; *Bobbala et al., 2018*), we fabricated MA-loaded BCN (MA-BCN) and found them to have the superior stimulatory capacity in vivo compared to MA-loaded poly(D,L-lactide-co-glycolide) (PLGA) nanocarriers (MA-PLGA), which is a widely used nanocarrier platform for Mtb vaccine development (*Lin et al., 2018*; *Khademi et al., 2018*). Furthermore, we found that after MA-BCN vaccination, MA could persist for 6 weeks post-vaccination within alveolar macrophages, a phenomenon which was dependent on the presence of an encapsulating vector, nanocarrier, or bacterial, but not the route of vaccination. Due to the enhanced stability of the BCN architecture that can support the loading of diverse and multiple payloads (*Bobbala et al., 2020*; *Modak et al., 2020*), we were able to efficiently co-encapsulate within BCN both MA and Ag85B, an immunodominant protein antigen of Mtb (*Huygen, 2014*). Interestingly, while both antigens activated their corresponding antigen-specific T cells, only MA resulted in antigen persistence, suggesting a potentially unique characteristic of subunit vaccines containing lipid antigens.

## Results

### BCN morphology preserved after MA loading

We scalably assembled spherical BCN with and without loaded MA using FNP as previously described (*Bobbala et al., 2018*). Dynamic light scattering (DLS) showed that both MA-BCN and BCN were consistent in size (364±19 nm and 354±14 nm, respectively), and were monodisperse based on their

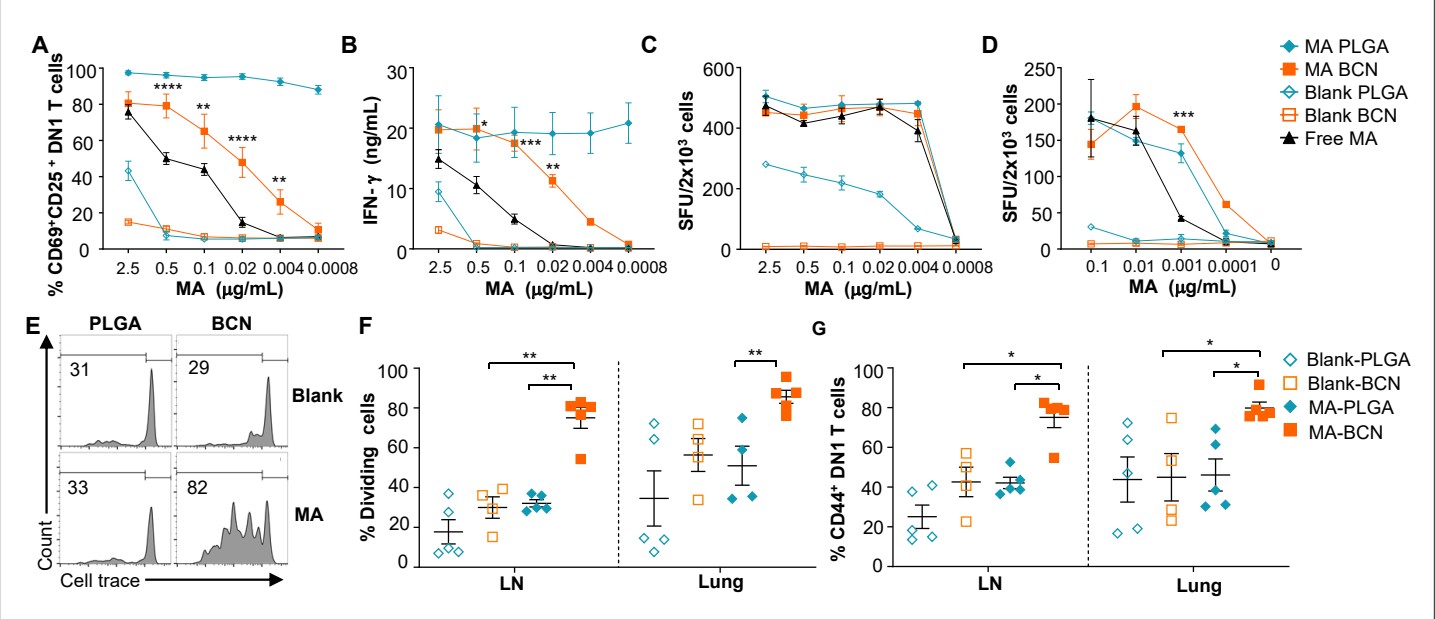

**Figure 2.** MA-loaded bicontinuous nanospheres (MA-BCN) effectively activated MA-specific T cells in vitro and in vivo. (**A, B**) Bone marrow-derived dendritic cells (BMDCs) were pulsed for 18 hr. with selected nanoparticles at various concentrations, co-cultured for 48 hr. with DN1 T cells, and T cell activation was measured. (**A**) Percentage of CD69 and CD25-expressing DN1 T cells. (**B**) IFN-γ production measured by enzyme-linked immunosorbent assay (ELISA). (N=3). (**C, D**) ELISPOT of IFN-γ from MA-specific human M11 T cells cultured for 16 hr with monocyte-derived dendritic cells with selected nanoparticles at high (**C**) and low (**D**) concentrations. Data are representative of two independent experiments and displayed as mean ± SEM. Statistical analysis: two-way ANOVA, significance designated for MA-BCN vs free MA. (**E–G**) hCD1Tg mice were IN vaccinated with selected nanoparticles and CellTrace Violet-labeled DN1 T cells were adoptively transferred the next day. After 1 week, DN1 T cell activation and proliferation were measured. (**E**) Representative FACS plots of DN1 T cells in the lymph nodes (LN). Percentage of proliferating (**F**) and CD44-expressing (**G**) DN1 T cells in the LN and lung. Data represented as mean ± SEM. Statistical analysis: two-way ANOVA. *p<0.05, **p<0.01, ***p<0.001, ****p<0.0001.

polydispersity indices (PDIs) of 0.24±0.04 and 0.21±0.05, respectively (*Figure 1A*). We next verified that the BCN formulation maintained its characteristic interconnected aqueous channels using cryogenic transmission electronic microscopy (cryo-TEM) (*Figure 1B*) and negative staining transmission electron microscopy (TEM) (*Figure 1C*). Using small angle X-ray scattering (SAXS) studies, we confirmed the internal cubic organization of BCN. Bragg peaks at the √2, √4, and √6 ratios show that the primitive type of cubic internal organization was preserved between MA-BCNs and BCNs (*Figure 1D*). Thus, MA encapsulation did not disturb the BCN architecture. We also manufactured a PLGA nanocarrier formulation (PLGA-NP) with and without MA encapsulation. PLGA-NP morphology was confirmed using cryo-TEM (*Figure 1—figure supplement 1A*) and SAXS (*Figure 1—figure supplement 1B*) and the size of blank PLGA-NP and MA-loaded PLGA-NP (MA-PLGA) were found to be 169±4 and 163±5 nm, respectively, with PDIs of 0.18±0.04 and 0.19±0.12, respectively (*Figure 1E*). Using the coumarin derivatization method (*Shang et al., 2018*), we found that BCNs could more efficiently encapsulate MA than PLGA-NPs (*Figure 1F*). We previously noted that the highly stable cubic architecture of BCN can lead to the retention of cargo within the endolysosomal compartment (*Bobbala et al., 2020*). Indeed, Texas Red-Dextran loaded BCN were mostly found in lysosomes while Texas Red-Dextran loaded PLGA-NP could be found both within the lysosome and cytosol of cells (*Figure 1—figure supplement 2A, B*).

## MA-BCN activates MA-specific T cells in vitro and in vivo

As PLGA has served as a component of a wide range of vaccine formulations, including Mtb vaccines (*Lin et al., 2018*; *Khademi et al., 2018*), we benchmarked PEG-*b*-PPS BCN against PLGA-NP as a nanocarrier platform for lipid antigens. Both MA-BCN and MA-PLGA were highly effective in activating both mouse (DN1) and human (M11) MA-specific T cells in vitro and inducing IFN-γ production (*Figure 2A–D*). In particular, MA-BCN were significantly better at activating MA-specific T cells compared to free MA at equivalent concentrations (*Figure 2A, B and D*). In comparison to MA-BCN,

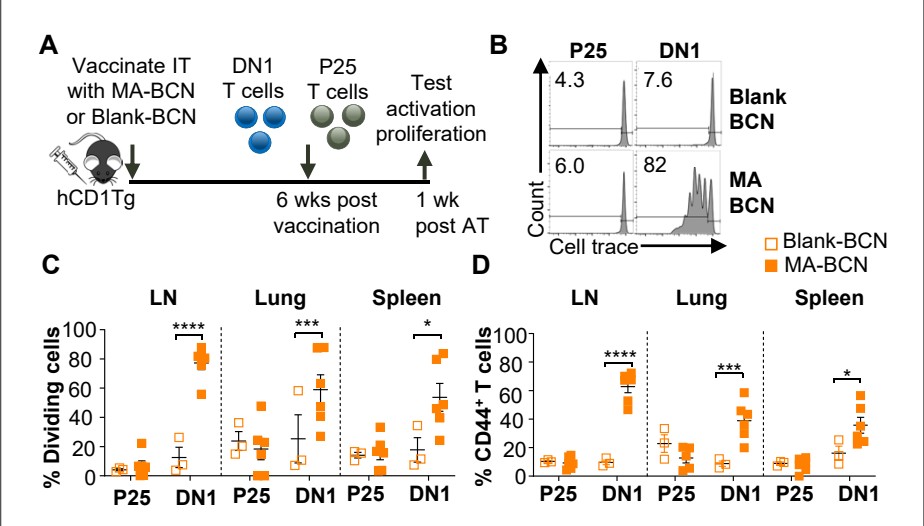

**Figure 3.** Vaccination with MA-bicontinuous nanospheres (MA-BCN) leads to antigen persistence 6 weeks post-vaccination. hCD1Tg mice were IT vaccinated with MA-BCN or BCN. 6 weeks later, CellTrace-labeled p25 and DN1 T cells were adoptively transferred, and T cell activation was measured after 1 week. (**A**) Experimental design. (**B**) Representative FACS plots of p25 and DN1 T cells in the lymph nodes (LN). Percentage of proliferating (**C**) and CD44-expressing (**D**) p25 and DN1 T cells in the LN, lung, and spleen. N=3 or 6 per condition. Outliers were identified through the Grubbs method with alpha = 0.05 with one sample removed. Data represented as mean ± SEM. Statistical analysis: two-way ANOVA. *p<0.05, **p<0.01, ***p<0.001, ****p<0.0001.

MA-PLGA was significantly more effective at stimulating mouse MA-specific T cells in vitro at lower concentrations. Interestingly, even in the absence of MA, PLGA-NP showed strong stimulation, particularly for M11 T cells. These results demonstrate that MA encapsulation within BCN enhances its ability to stimulate antigen-specific T cells and reveal a considerable background, and thus difficult to control, the stimulatory effect of blank PLGA-NP. In contrast, blank BCN showed little background immunostimulation (*Burke et al., 2022*), allowing a more controlled dose-dependent increase in T cell responses as the loaded MA concentration increased.

We next tested the ability of MA-loaded and unloaded BCN and PLGA to stimulate DN1 T cells in vivo. Intranasal (IN) vaccination of hCD1Tg mice was followed by the adoptive transfer of DN1 T cells. Surprisingly, we found that vaccination with MA-BCN induced a significantly higher percentage of proliferation and activation in DN1 T cells than MA-PLGA in the draining lymph nodes (LN) and lungs at 1 week post-vaccination (*Figure 2E–G*). The extent of cell proliferation induced by MA-PLGA was not significantly different from that of blank PLGA (*Figure 2F*). Since MA-BCN effectively stimulated MA-specific T cells in vivo while MA-PLGA did not, we focused on further characterizing and assessing vaccination via solely MA-BCN formulations.

## MA persists while Ag85B does not after vaccination with Ag85B-MA-BCN

Given the lack of knowledge regarding BCN and lipid antigen kinetics, it would be of interest to test whether delivered antigen could persist in the mouse weeks post-vaccination (*Figure 3A*). We switched from intranasal vaccination to intratracheal (IT) vaccination, as it allowed for the reliable delivery of a larger volume of the vaccine. We found that a significant proportion of the adoptively transferred DN1 T cells could proliferate and become activated in the LNs, lung, and spleen of MA-BCN-vaccinated mice 6 weeks post-vaccination (*Figure 3B–D*). This activation and proliferation were MA-specific, as no activation and proliferation of Ag85B-specific p25 Tg (p25) T cells were observed (*Tamura et al., 2004*).

Since BCN allows for the co-loading of lipid and protein antigens, we assessed the ability of dual-loaded Ag85B-MA-BCN to activate antigen-specific T cells as well as lead to antigen persistence. We co-loaded Ag85B and MA in BCN and found Ag85B-MA-BCN to have a diameter of 380 nm and PDI of 0.223, comparable to MA-BCN (*Figure 4A*). Encapsulation efficiency was also relatively high at

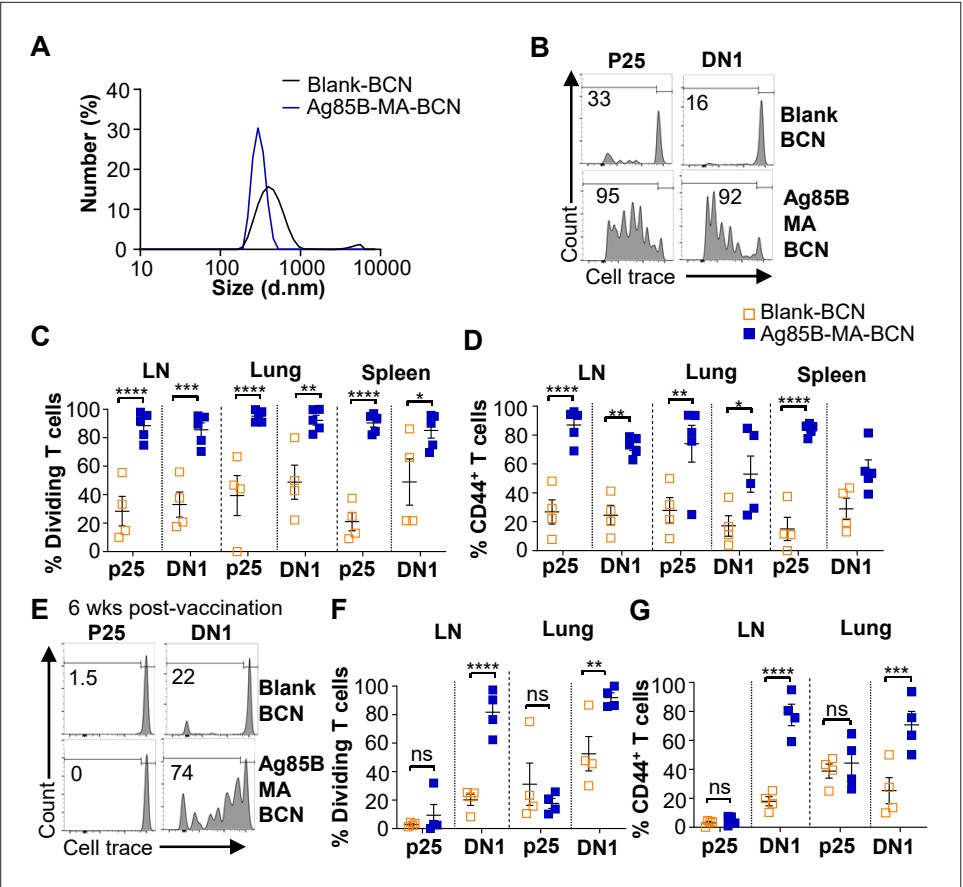

**Figure 4.** Ag85B and Mycolic acid (MA) dual-loaded bicontinuous nanospheres (BCNs) activate antigen-specific T cells without Ag85B persistence. (**A**) Dynamic light scattering (DLS) analysis of blank BCN and Ag85B-MA-BCN. (**B–D**) hCD1Tg mice were IT vaccinated with Ag85B-MA-BCN or BCN and CellTrace-labeled p25 and DN1 T cells were adoptively transferred the next day. After 1 week, T cell activation and proliferation were measured. (**B**) Representative FACS plots of p25 and DN1 T cells in the lymph nodes (LN). Percentage of proliferating (**C**) and CD44-expressing (**D**) p25 and DN1 T cells in the LN, lung, and spleen (N=4 or 5). (**E–G**) hCD1Tg mice were IT vaccinated with blank BCN or Ag85B-MA-BCN. 6 weeks later, CellTrace-labeled p25 and DN1 T cells were adoptively transferred, and T cell activation was measured after 1 week. (**E**) Representative FACS plots of p25 and DN1 T cells in the LN. (**F**) Percentage of proliferating p25 and DN1 T cells. (**G**) Percentage of CD44-expressing p25 and DN1 T cells in the LN and lung. (N=4). Data represented as mean ± SEM. Statistical analysis: two-way ANOVA. ns = not significant, *$p < 0.05$, **$p < 0.01$, ***$p < 0.001$, ****$p < 0.0001$.

70%. Vaccination of hCD1Tg mice with Ag85B-MA-BCN activated and induced proliferation of both p25 and DN1 T cells in the LN, lung, and spleen (*Figure 4B–D*) at 1 week post-vaccination. In addition, DN1 T cells were still able to be activated 6 weeks post-vaccination in the context of an Ag85B-MA-BCN vaccination, while p25 T cells did not show increased activation or proliferation compared to blank BCN control at this time point (*Figure 4E–H*). Therefore, MA, but not Ag85B, appears to persist in the context of an Ag85B-MA-BCN vaccination.

## MA persistence was dependent on encapsulation but not route of delivery or delivery vector

To determine whether BCN encapsulation contributed to MA persistence, we designed an experiment that would allow direct comparison between MA-BCN and free MA. To account for their differential efficiency in stimulating DN1 T cells, we pulsed bone marrow-derived dendritic cells (BMDCs) from hCD1Tg mice with MA or MA-BCN at 10 µg/mL and 5 µg/mL MA concentration, respectively (*Figure 5A*). hCD1Tg mice were immunized with MA or MA-BCN-pulsed BMDCs at either 1 week or 6 weeks before the adoptive transfer of DN1 T cells, allowing comparison of short-term *vs.* long-term

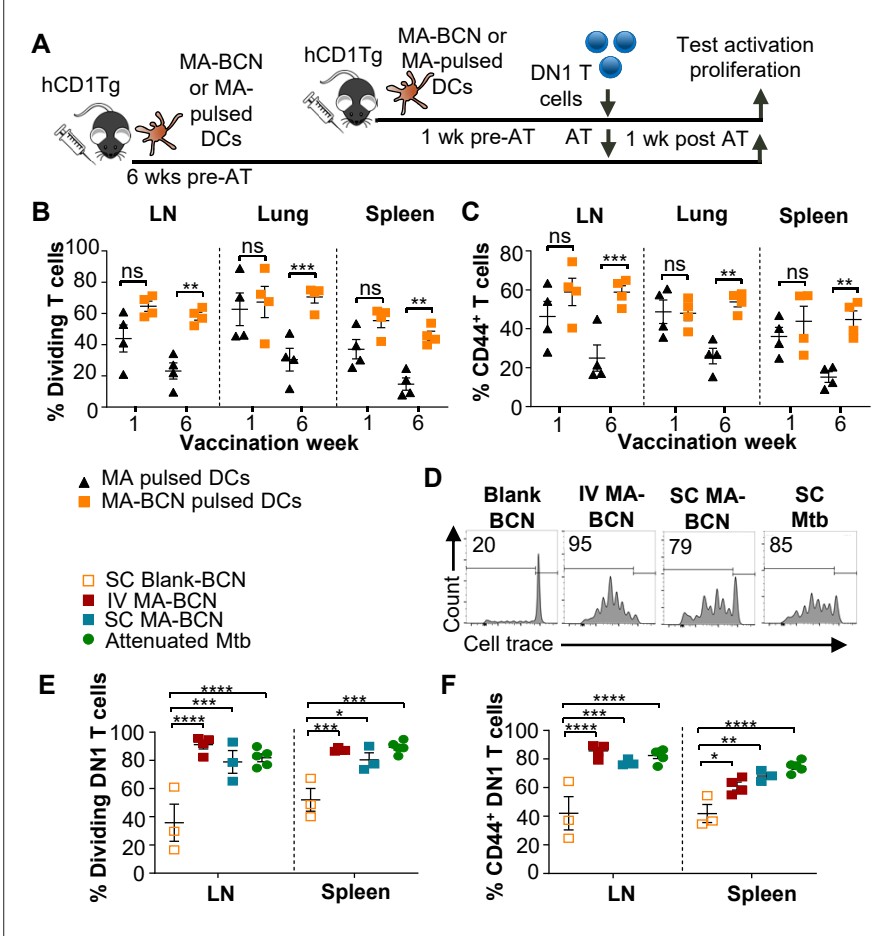

**Figure 5.** Route of vaccination or vector type does not affect antigen persistence. hCD1Tg BMDCs were pulsed with MA-bicontinuous nanospheres (MA-BCN) or mycolic acid (MA) at concentrations of 5 mg/mL and 10 mg/mL, respectively. hCD1Tg mice were IT vaccinated with MA or MA-BCN pulsed bone marrow-derived dendritic cells (BMDCs) at 6 weeks or 1 week prior to the adoptive transfer of CellTrace-labeled DN1 T cells. T cell activation and proliferation were measured 1 week after adoptive transfer. (**A**) Experimental design. (**B, C**) Percentage of proliferating (**B**) and CD44-expressing (**C**) DN1 T cells in the lymph nodes (LN), lung, and spleen of vaccinated mice. N=4 per condition. hCD1Tg mice were vaccinated SC (subcutaneously) or IV (intravenously) with blank BCN, MA-BCN, or attenuated *Mycobacterium tuberculosis* (Mtb) strain and 6 weeks later CellTrace-labeled DN1 T cells were adoptively transferred. (**D**) Representative FACS plots of DN1 T cells in the LN of mice vaccinated with indicated conditions. (**E**) Percentage of proliferating DN1 T cells and (**F**) Percentage of CD44-expressing DN1 T cells in the LN and spleen of mice vaccinated with Blank-BCN (SC), MA-BCN (IV and SC), and attenuated Mtb. N=3–5 per condition. Data represented as mean ± SEM. Statistical analysis: two-way ANOVA. ns = not significant, *p<0.05, **p<0.01, ***p<0.001, ****p<0.0001.

The online version of this article includes the following figure supplement(s) for figure 5:

**Figure supplement 1.** Mycolic acid (MA) encapsulated by either bicontinuous nanospheres (BCN) or MC formulation leads to lipid antigen persistence.

**Figure supplement 2.** P25-specific T cell activation and proliferation and bacterial burden in attenuated *Mycobacterium tuberculosis* (Mtb) vaccination.

vaccination conditions. We found that while there were no significant differences in DN1 T cell activation and proliferation between mice vaccinated with free MA and MA-BCN pulsed BMDCs at 1 week post-vaccination, after 6 weeks, DN1 T cell response could only be detected in the MA-BCN vaccination condition (*Figure 5B, C*). Thus, BCN encapsulation contributed to the persistence of MA. We then tested whether the BCN structure could contribute to antigen persistence by comparing PEG-*b*-PPS BCNs to PEG-*b*-PPS MCs using a similar experimental setup (*Figure 5—figure supplement*

*1A*). We found that MA-BCN and MA-MC overall had similar ability to activate DN1 T cells in both short-term and long-term vaccination time points (*Figure 5—figure supplement 1B, C*). Thus, the structure of the encapsulating NP did not play a significant role in the antigen persistence of MA.

To investigate if the IT route of vaccination played a role in the development of antigen persistence, we tested intravenous (IV) and subcutaneous (SC) vaccination routes with MA-BCN. DN1 T cell activation and proliferation were detected at 6 weeks post-vaccination through both routes of vaccination (*Figure 5D–F*), suggesting that the antigen persistence is not unique to IT vaccination. To assess whether Mtb lipid antigen persistence could also be observed in mice vaccinated with an attenuated strain of Mtb, we vaccinated hCD1Tg mice SC with Mtb mc$^2$6206 strain and tested for the persistence of both Ag85B and MA after 6 weeks. The Mtb mc$^2$6206 strain displays less virulence in mouse studies than BCG but still confers long-term protection against virulent Mtb challenge equivalent to BCG (*Sampson et al., 2004*). We confirmed that no bacterial burden remained at the 6 weeks time point (*Figure 5—figure supplement 2A*). Similar to MA-BCN-vaccinated mice, DN1 T cells were activated and proliferated in the draining lymph nodes and spleen of mice vaccinated with the attenuated Mtb strain (*Figure 5D–F*). In line with our findings with BCN, no proliferation was seen in p25 T cells (*Figure 5—figure supplement 2B–D*). These results suggest that the incorporation of MA in a nano-carrier vaccine allows for the mimicry of the lipid antigen persistence induced by vaccination with an attenuated Mtb vaccine.

## MA persists within alveolar macrophages in the lung

To identify the location of antigen persistence, we determined the biodistribution of BCN after IT vaccination using BCN loaded with the hydrophobic dye DiD (DiD-BCN). Since DiD is a lipophilic dye which, like MA, stably loads into the BCN bilayers, we could track the localization of BCN without having to perform additional chemical modifications. We found that IT-administered DiD-BCN were located almost exclusively within the lung at all time points tested (4 hr, 24 hr, 48 hr, 6 days, and 6 weeks) (*Figure 6A*). We were able to detect fluorescence through both in vivo imaging system (IVIS) and flow cytometry at 6 weeks post-vaccination that was significantly above unvaccinated control (*Figure 6A, B*), despite DiD quickly losing fluorescence upon mixture into aqueous environments (*Invitrogen, 2023*), suggesting that some DiD-BCN structure remained intact at this time point. Flow cytometric analysis (*Figure 6—figure supplement 1*) revealed that DiD BCN was primarily found within alveolar macrophages (AMs) at both early and late time points with neutrophils and DCs showing some fluorescence at early time points (4 and 24 hr) (*Figure 6C*).

We next tested whether MA likewise persisted primarily within AMs. 6 weeks after vaccination, we enriched for AMs using a column-based magnetic cell isolation system containing SiglecF antibody, which is highly expressed in AMs and a small population of eosinophils in the lung (*Figure 6D*). We co-cultured enriched AMs or the flow through with or without hCD1Tg-expressing BMDCs and DN1 T cells (*Figure 6—figure supplement 1*). While both enriched AM and flow-through fractions from MA-BCN vaccinated mice co-culture with BMDCs could activate DN1 T cells (*Figure 6E*), co-culture with enriched AMs led to significantly higher DN1 T cell activation, suggesting AMs are the primary location of MA persistence. The ability of flow through to likewise lead to DN1 T cell activation suggests MA may also persist in other cell types or could be attributed to residual AMs in flow through. Furthermore, AMs could not themselves present MA to DN1 T cells, noted by the lack of T cell activation in the absence of BMDCs, suggesting that MA may be transferred from AMs to DCs for presentation. In fact, CD1b is not expressed on AMs (*Felio et al., 2009*; *Barral and Brenner, 2007*) and, therefore, the presence of CD1b-expressing DCs may be necessary both in vitro and in vivo for T cell activation to occur. These data indicate that MA-BCN mainly persists within AMs in the lung after IT vaccination and CD1b-expressing DCs are required for antigen presentation and activation of MA-specific T cells.

## Vaccination leads to DN1 T cells differentiating into T follicular helper-like T cells

To study the memory phenotype of DN1 T cells after MA-BCN vaccination, we constructed a mixed DN1 bone marrow (BM) chimera mouse model (DN1-hCD1Tg), since adoptively transferred DN1 T cells in unvaccinated mice cannot survive long-term (*Figure 7A*). After vaccination, the percent of CD44$^+$ DN1 T cells in the blood increased quickly and remained high until the last time point

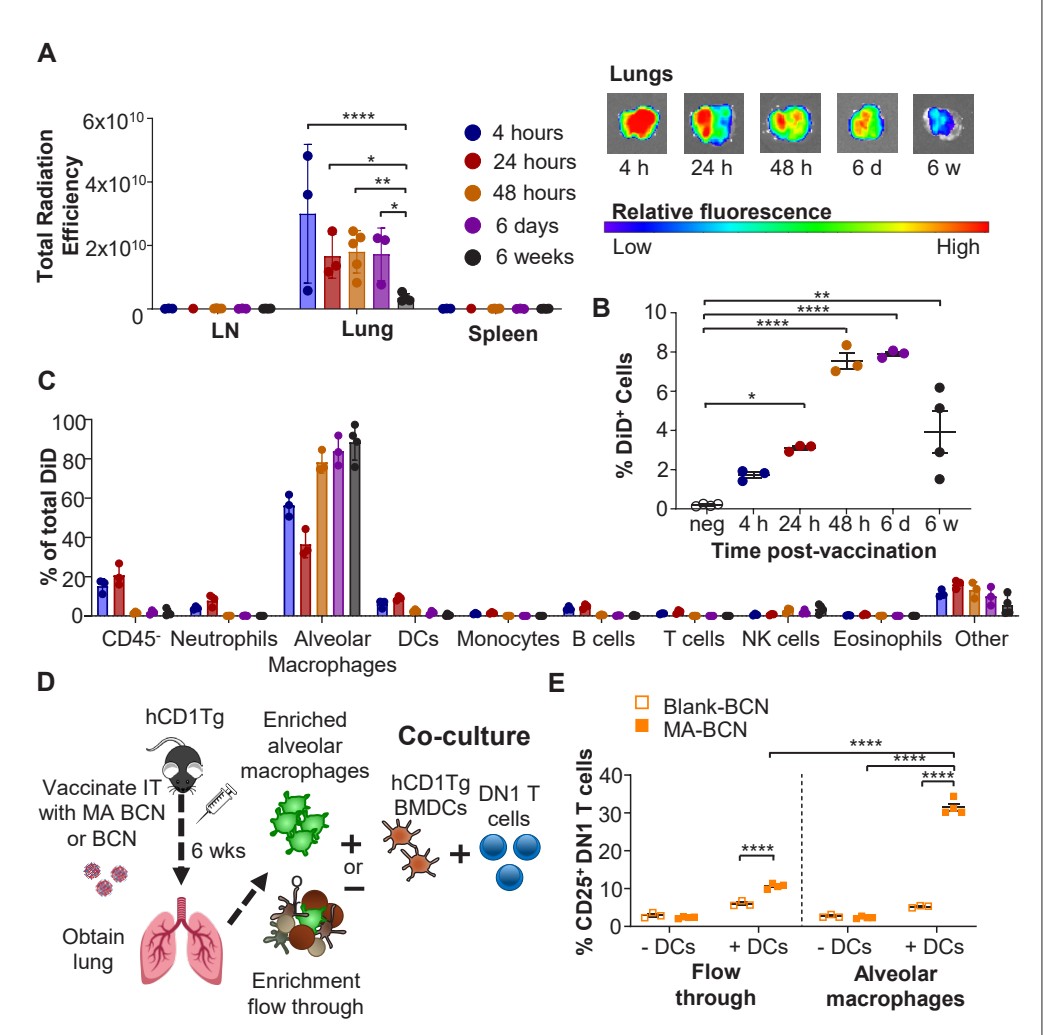

**Figure 6.** Persistent mycolic acid (MA) remains in part due to encapsulation inside alveolar macrophages and activates T cells through DCs. B6 or hCD1Tg mice were IT vaccinated with bicontinuous nanospheres (BCN) loaded with a hydrophobic fluorescent dye (DiD) at 6 weeks, 6 days, 48 hr, 24 hr, or 4 hr prior to the experiment. Lymph nodes (LN), lung, and spleen were analyzed using (**A**) In Vivo Imaging System (IVIS). Single-cell suspension was then obtained and presence of DiD BCN was quantified in (**B**) total lung, and (**C**) CD45$^-$ cells, neutrophils (Ly6G$^+$), alveolar macrophages (CD11c$^+$SiglecF$^+$), DCs (CD11c$^+$), monocytes (CD11b$^+$CD11c$^-$), B cells (CD19$^+$), T cells (CD3$^+$), NK cells (NK1.1$^+$), eosinophils (CD11c$^-$SiglecF$^+$) by flow cytometry. hCD1Tg mice were IT vaccinated with either BCN or MA-BCN. After 6 weeks, alveolar macrophages were enriched from the lungs using anti-SiglecF. Enriched or flow-through cells were co-cultured with DN1 T cells in the presence or absence of hCD1Tg-expressing bone marrow-derived dendritic cells (BMDCs) for 48 hr, and DN1 T cells activation was measured. (**D**) Experimental design. (**E**) Percentage of CD25-expressing DN1 T cells. Data represented as mean ± SEM. Statistical analysis: two-way ANOVA. ns = not significant, *p<0.05, **p<0.01, ***p<0.001.

The online version of this article includes the following figure supplement(s) for figure 6:

**Figure supplement 1.** Gating strategy for analyzing biodistribution of bicontinuous nanospheres (BCN) in the lung.

**Figure supplement 2.** Representative FACS plot of alveolar macrophages within enriched and flow through fractions.

---

of 40 days (*Figure 7B*). Various classical memory T cell subsets have been characterized 1 month after the initial antigen encounter, we, therefore, determined the DN1 T cell phenotype 6 weeks post-vaccination. We found that the most prevalent memory population within DN1 T cells were CD44$^+$CD62L$^+$, markers used to define central memory T cells, particularly in the LN (*Figure 7C* and

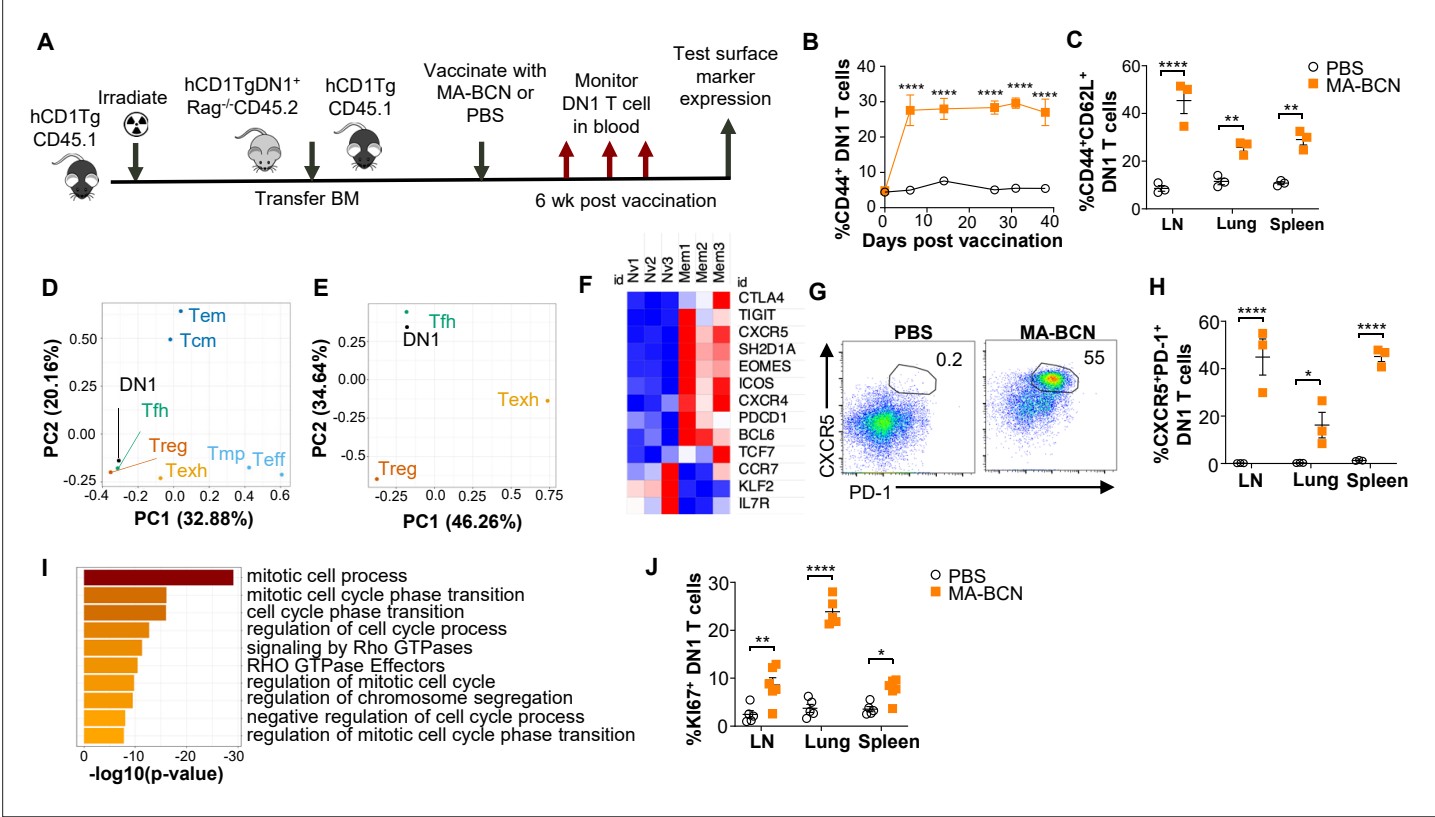

**Figure 7.** MA-bicontinuous nanospheres (MA-BCN) vaccination leads to DN1 T cell differentiation into T follicular helper-like T cells. Mixed bone marrow chimeric mice were created by adoptive transfer of DN1 bone marrow and congenic CD45.1 hCD1Tg bone marrow to irradiated hCD1Tg mice. After 5weeks, mice were vaccinated with either MA-BCN or PBS, and DN1 surface marker expression was monitored in the blood and in organs at 6weeks. (**A**) Experimental design. (**B**) Percent CD44+ DN1 T cells in the blood at various time points post-vaccination. (**C**) Percent CD44+CD62L+ DN1 T cells in indicated organs at 6weeks post-vaccination. Memory (CD44+CD62L+) and naïve (CD44-CD62L+) DN1 T cells were sorted from MA-BCN vaccinated hCD1Tg-DN1 BM chimeras at 6weeks post-vaccination and subjected to RNA-seq analysis. N=3 per condition. (**D, E**) PCA of log fold change of gene expression for each T cell subset was performed relative to internal naïve T cell control. (**F**) Relative expression of key T_FH cell and differentially expressed genes (DEGs) in memory subset. (**G**) Representative FACS plots of DN1 T cells in the lymph nodes (LN) of mice. Percentage of (**H**) CXCR5+PD1+ or (**J**) KI67+ DN1 T cells DN1 T cells in LN, lung, and spleen. N=3–5 per condition. (**I**) Gene enrichment analysis of upregulated DEGs was performed using Metascape. Data represented as mean ± SEM. Statistical analysis: two-way ANOVA. *p<0.05, **p<0.01, ****p<0.0001.

The online version of this article includes the following figure supplement(s) for figure 7:

**Figure supplement 1.** Memory and naïve DN1 T cells display distinct gene expression profiles.

*Figure 7—figure supplement 1A*). To characterize the memory DN1 T cells, we performed RNA-seq analysis on sorted CD44+CD62L+ (memory) and CD44-CD62L+ (naïve) DN1 T cells from LNs of MA-BCN vaccinated DN1-hCD1Tg BM chimeric mice. We found that memory and naïve DN1 T cells clustered separately after principal component analysis (PCA), despite these samples coming from the same animals (*Figure 7—figure supplement 1B*). A total of 995 differentially expressed genes (DEGs) were identified of which 542 upregulated and 453 downregulated in the memory subset (*Figure 7—figure supplement 1C*). Next, we determined which T cell population memory DN1 T cells most resembled. Toward this end, we obtained data from the ImmGen database (including terminally differentiated effector (Te), memory precursor (Tmp), central memory (Tcm), effector memory (Tem), regulatory (Treg))[42], and two additional publications (follicular helper (Tfh)[43], exhausted (Texh)[44]) and compared the respective DEG lists. Using PCA, we found that memory DN1 T cells clustered most closely to Tfh, Treg, and Texh (*Figure 7D*) and when this subset was isolated, most closely to Tfh cells (*Figure 7E*).

Within the DEGs, we noted upregulation of key Tfh cell transcription factors BCL6 and TCF1 (*Tcf7*), chemokines CXCR5 and CXCR4, and surface receptor ICOS, important for Tfh cell migration into the germinal center, and adaptor protein SAP (*Sh2d1a*), important for T cell-dependent B cell immunity (*Figure 7F*; *Ma et al., 2021*). Inhibitory receptors PD-1 (*Pdcd1*), TIGIT, CTLA4, and transcription

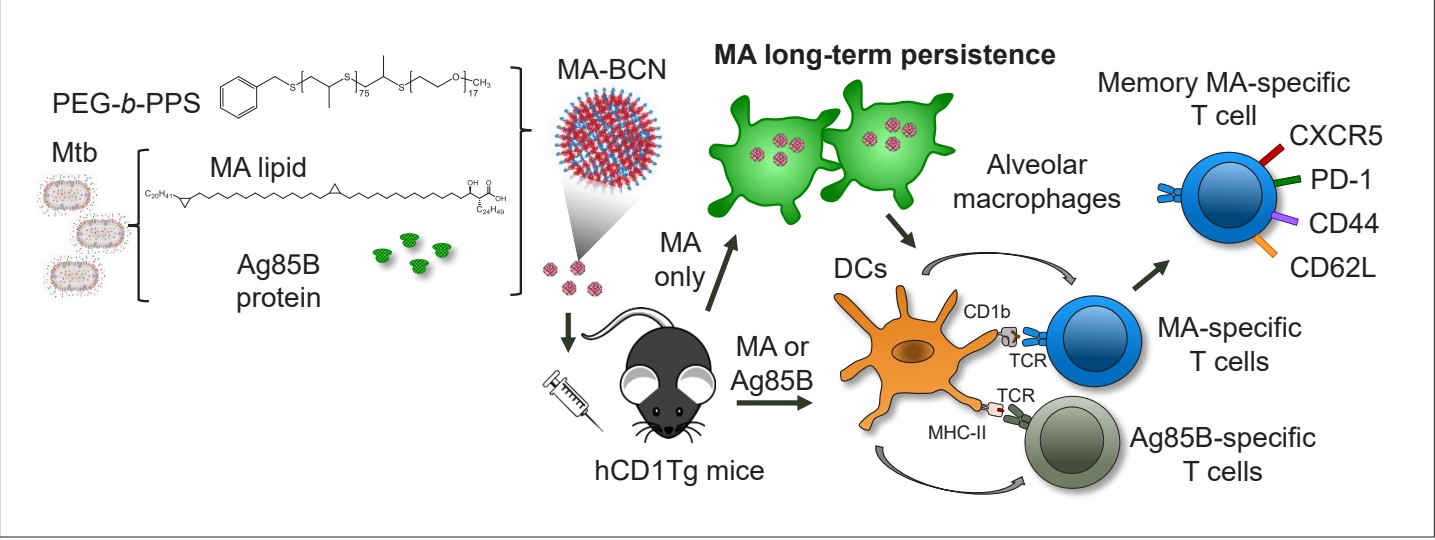

**Figure 8.** Summary schematic illustrates MA antigen persistence and memory differentiation of antigen-specific T cells induced by vaccination with MA-loaded nanoparticle vaccine. *Mycobacterium tuberculosis* (Mtb) derived mycolic acid (MA) lipid and Ag85B protein are encapsulated into bicontinuous nanosphere (BCNs) composed of PEG-*b*-PPS polymer. These MA-BCN or Ag85B-MA-BCN are used for intratracheal (IT) vaccination of human CD1 transgenic (hCD1Tg) mice. Following vaccination, dendritic cells (DCs) present the antigens, leading to the activation of CD1b-restricted MA-specific or MHCII-restricted Ag85B-specific T cells (DN1 or p25-specific T cells, respectively). Notably, MA can persist within alveolar macrophages and continue to activate MA-specific T cells even at 6 weeks post-vaccination. These MA-specific T cells have the capacity to differentiate into memory T cells with a T follicular-like phenotype.

factor EOMES, which are known to be expressed on Tfh cells were also upregulated. In addition, we observed downregulation of transcription factor KLF2, which can inhibit Tfh cell differentiation, and CCR7 and IL-7R, receptors known to have decreased expression in $T_{FH}$ cells. Consistent with our transcriptome analysis, we found an increased percentage of CXCR5+PD-1+ DN1 T cells in MA-BCN vaccinated mice 6 weeks post-immunization (*Figure 7G, H*). Both our transcriptomic and experimental data highlighted that memory, but not naïve DN1 T cells, were proliferating, given the enrichment for the mitotic cell cycle process (*Figure 7I*) and increased KI67+ percentage within this population (*Figure 7J*). MA-BCN vaccination thus led to the differentiation of DN1 T cells into a proliferating Tfh-like T cell population.

## Discussion

Research on the development of subunit vaccines containing lipid antigens is limited. This is in part due to a lack of research into appropriate lipid vaccine formulations and their associated properties. Some early evidence suggests that lipid vaccines could contribute to protection in the Mtb challenge in the guinea pig model (*Dascher et al., 2003*; *Larrouy-Maumus et al., 2017*), which naturally express group 1 CD1 molecules (*Eckhardt and Bastian, 2021*). Vaccination with a formulation composed of total Mtb lipid encapsulated within liposomes with either dimethyl-dioctadecyl-ammonium (DDA) and/or QS-21 adjuvants showed lower bacterial burden and pathology in lung and spleen compared to unvaccinated controls (*Dascher et al., 2003*). In another study, two Mtb lipids (diacylated sulfoglycolipids and phosphatidylinositol dimannosides) encapsulated into liposomes formulated with DDA and trehalose 6,6'-dibehenateled (TDM) led to a decreased bacterial burden in the spleen and overall improved pathology in the lung and spleen (*Larrouy-Maumus et al., 2017*). This work motivated our current study, where we used hCD1Tg mice that express human group 1 CD1 molecules, including CD1b and MA-specific (DN1), as well as p25-specific T cells to investigate the use of nanocarriers coloaded with both Mtb lipid and protein antigens during vaccination (*Figure 8*).

We found that PEG-*b*-PPS BCN could effectively encapsulate MA as well as the protein antigen Ag85B to elicit activation of antigen-specific T cells in vaccinated mice (*Figure 4*). To our knowledge, this is the first time a dual protein and lipid loaded nanocarrier has been shown to effectively stimulate both lipid and peptide antigen-specific T cells simultaneously in vivo. PEG-*b*-PPS BCN can

encapsulate both hydrophobic and hydrophilic molecules, making it an excellent platform for subunit vaccine delivery as both antigens and adjuvants can be co-loaded and delivered to the same cellular targets. Prior work on TB vaccines has shown that the inclusion of several protein antigens spanning the Mtb life cycle along with adjuvants with multiple boosts is paramount in the development of a vaccine that can provide effective protection against Mtb (*Lin et al., 2018*; *Schrager et al., 2020*). As we have established that PEG-*b*-PPS BCN can easily incorporate a variety of payloads, future work will involve determining the optimal antigen and adjuvant combination for providing protection in a Mtb challenge experiment.

As several formulations of PLGA-NP have been approved for clinical uses (*Wang et al., 2021*), we compared MA encapsulated by either BCN or PLGA-NP and found that while PLGA appeared to offer greater, yet less controllable stimulation in vitro (*Figure 2*), BCN exhibited significantly greater efficacy in vivo (*Figure 2*). The result is not completely unexpected as a more fragile PLGA-NP could allow for fast intracellular release and antigen presentation in vitro while also leading to significant antigen loss prior to APC interaction in vivo (*Figure 3*; *Ding and Zhu, 2018*). Previous studies have documented PLGA's innate pro-inflammatory effects, while other studies have highlighted the absence and suppression of inflammatory response (*Barillet et al., 2019*; *Ma et al., 2021*; *Nicolete et al., 2011*). As PLGA has not been previously used for lipid antigen delivery via IT vaccination, our findings suggest that PLGA may have limited effectiveness in such a vaccine system, although alternative formulations or polymer compositions could potentially yield improved results. In contrast and as previously observed for PEG-*b*-PPS nanocarriers (*Burke et al., 2022*), we showed that unloaded BCN have little inflammatory effect (*Figure 2*), allowing their immunomodulation to be dictated by the properties of the loaded antigen and adjuvant. Thus, PEG-*b*-PPS is a promising polymer for the formulation of subunit vaccines.

We proceeded with a pulmonary (IN or IT) vaccination approach in this study based on emerging evidence that mucosal vaccination can induce a stronger immune response in the lung against Mtb in humans (*Stylianou et al., 2019*) and improve protection following vaccination with nanocarriers in mice (*Ballester et al., 2011*). We found that in the pulmonary route of vaccination, BCN primarily was taken up and remained within alveolar macrophages in the lung (*Figure 6*). In fact, the ability of nanocarriers to target alveolar macrophages or other APCs may contribute to their distinct capacity for inducing antigen persistence, which was not seen with cellular carriers such as MA-pulsed BMDCs (*Figure 5*). Alveolar macrophages are the first line of defense of the lungs, responsible for keeping alveoli sterile by taking up inhaled particles and pathogens through phagocytosis (*Fels and Cohn, 1986*; *Bowden, 1984*; *Geiser, 2010*). Targeting these cells may be particularly beneficial in a TB vaccine design since they may also be the main site of Mtb latency (*Pai et al., 2016*). While MA remained within alveolar macrophages in the lung, T cell activation appeared to occur prominently within the draining lymph nodes (*Figure 2*), suggesting that either MA was transported from the lungs to lymph nodes where the presentation took place or T cells activated in the lung migrated to the lymph nodes. The first option seems most likely, as CD1b-expressing DCs are more prevalent in the lymph nodes (*Dougan et al., 2007*). Furthermore, in Mtb infection models, it has been previously shown that apoptotic macrophages and vesicles could deliver mycobacterial antigens to DCs that then activate cognate T cells (*Espinosa-Cueto et al., 2017*; *Schaible et al., 2003*). Similarly, we showed that alveolar macrophages from MA-BCN vaccinated mice could only lead to T cell activation in the presence of BMDCs from hCD1Tg mice (*Figure 6*).

We demonstrated MA persistence in the lung 6 weeks post-vaccination using an in vivo antigen presentation assay with adoptive transfer of naïve MA-specific T cells. Sustained release of antigen through vaccination is thought to be key in the development of a robust adaptive immune response (*Silva et al., 2016*; *Correia-Pinto et al., 2013*). Recently, spike protein was found within the draining lymph nodes of vaccinated human subjects 60 days after COVID-19 vaccination with mRNA-1273 or BNT162b2 (*Röltgen et al., 2022*). In mice, pulmonary vaccination with adenovirus expressing influenza nucleoprotein supported long-term maintenance of CD8 $T_{RM}$ (resident memory) cells (*Uddbäck et al., 2021*). Unlike in this work, antigen persistence did not occur after subcutaneous vaccination, suggesting a distinct mechanism of antigen archiving. In a different vaccine model, subcutaneous administration of ovalbumin paired with adjuvants polyI:C and anti-CD40 led to antigen archiving by LECs (*Tamburini et al., 2014*; *Walsh et al., 2021*). Similar to our findings, antigen-specific T cell activation required cross-presentation by DCs (*Tamburini et al., 2014*; *Kedl et al., 2017*). Both LECs and

FDCs retain antigens within a non-degradative compartment (*Tamburini et al., 2014*; *Heesters et al., 2013*), suggesting that residual antigens in our model may similarly remain in the endosome within undigested BCN. We have previously demonstrated the enhanced stability of BCN within cell endosomes in vitro for extensive periods of time (*Bobbala et al., 2020*). This possibility is corroborated by the finding that DiD, which is easily quenched in aqueous solution (*Invitrogen, 2023*), remained fluorescent 6 weeks post-vaccination (*Figure 6*). The lack of persistence for Ag85B could be due to its leakage out from the aqueous pores of BCN, which we have observed for other proteins (*Bobbala et al., 2018*). Peptide antigens may be modified to include a hydrophobic tail for stable retention within BCN and, therefore, allow for peptide antigen persistence as well. MA's extreme hydrophobic surface, which correlates with its low permeability as a part of the Mtb cell wall, may also contribute to its prolonged persistence in vivo (*Daffé et al., 2019*).

After infection and vaccination, peptide antigen persistence has been shown to occur through archiving by binding to CD21 receptors on follicular dendritic cells (FDCs) in lymph nodes (*Heesters et al., 2014*; *Hirosue and Dubrot, 2015*). MA antigen persistence may likewise be a natural phenomenon that can occur after infections, although its detection may be difficult given the lack of available assays to detect very small amounts of lipid antigen. T cells may, in fact, be more sensitive than any commercially available tool. While we determined that MA persists within AMs following pulmonary vaccination, this localization is likely dependent on the vaccination route. In the case of SC and IV routes, significant exposure of AMs to MA-BCN is unlikely to occur. Consequently, the exact localization of MA after vaccination via SC, IV, or attenuated Mtb remains to be determined, and it is plausible that other types of macrophages may be involved in this process. Additionally, we found that different forms of encapsulation (such as BCN, MC, and attenuate Mtb) can induce antigen persistence, DC internalization alone was not sufficient to replicate this phenomenon (*Figure 5*). Thus, the question still remains regarding the necessary properties of the delivery vector for the persistence of antigens to occur. Overall, MA antigen persistence is a robust phenomenon which occurs independently of the vector and vaccine route of administration, and thus may play a role in the efficacy of a variety of vaccine strategies. Additional work will focus on whether lipid antigen persistence is found across the lipid family or specific to lipids with certain properties.

Antigen persistence can also occur in chronic infections in which it is known to induce T cell exhaustion (*Barber et al., 2006*). Despite differentiating in an environment of persistent antigen exposure, memory DN1 T cells exhibited a gene expression profile more similar to that of Tfh cells rather than exhausted T cells (*Figure 7*). In particular, memory DN1 T cells upregulated the expression of TCF7 and BCL6 which are known to sustain a non-exhausted phenotype in CD8 T cells during chronic viral infections (*Wu et al., 2016*; *Wang et al., 2019*). Previous studies on antigen archiving have suggested that the mere persistence of antigen is insufficient for exhaustion to develop, and the inflammatory context likely plays a role in T cell differentiation (*Tamburini et al., 2014*). In many subunit vaccines, virulence factors are used as antigens to target and suppress the pathogen's capacity to infect the host. The persistence of lipid virulence factors could raise concerns regarding the safety of lipid vaccines. However, it is worth noting that otherwise toxic compounds such as the lipid TDM have been successfully incorporated into numerous vaccines without any known safety concerns to date (*Kwon et al., 2023*; *Lima et al., 2003*; *Niu et al., 2011*; *Sarkar et al., 2023*; *Welsh et al., 2013*; *Xin et al., 2013*). The absence of reported safety issues may be attributed to limited research or the administration of low doses of TDM. It is important to highlight that while MA plays a role in immune evasion, MA without additional functional groups has not been shown to elicit similar toxic effects as TDM (*Daffé et al., 2019*).

The use of memory T cells in the setting of antigen persistence may be debatable, given that memory response has historically referred to the adaptive immune response in the setting of antigen clearance (*Martin and Badovinac, 2018*). However, given that the persistence of MA occurs even in the setting of attenuated Mtb vaccination, Mtb lipid-specific long-term adaptive immunity will likely naturally exist in an antigen-persistent environment. We found that DN1 T cells in this setting expressed key markers of Tfh cells such as CXCR5, PD-1, BCL6, and TCF1 (*Figure 7*). T cells differentiate into Tfh cells through the expression of master transcription factor regulator BCL6, which directs the upregulation of chemokine receptors CXCR5 and CXCR4 and thus migration into the germinal center (*Johnston et al., 2009*; *Nurieva et al., 2009*). In the germinal center, Tfh cells provide help to B cells for affinity maturation and class switching. Other unconventional T cells are known to behave

similarly such as CD1d-restricted NKT cells, which can acquire a Tfh phenotype and provide B cell help through both cognate and non-cognate interactions (*Clerici et al., 2015*). Additional work will look at whether DN1 or other group 1 CD1-restricted T cells can functionally act as Tfh cells to provide B cell help, as may be suggested by the existence of IgG antibodies against MA and glycolipid lipoarabinomannan in TB patients (*Correia-Neves et al., 2019*; *Schleicher et al., 2002*; *Pan et al., 1999*).

While the persistence of protein antigen has been shown to improve the quality of protective immunity (*Demento et al., 2012*; *Tamburini et al., 2014*; *Bioley et al., 2015*; *Silva et al., 2013*), it is not yet known how lipid antigen persistence may affect the protective efficacy of lipid-specific T cells. At the very least, antigen persistence increases the probability of T cell/DC encounters by increasing the time that the antigen is available. Additional work will need to address the function of lipid-specific T cells differentiated in antigen-persistent environments. As the ultimate goal of this work is to improve the efficacy of existing subunit vaccines, future work will assess the protective capacity of lipid antigen vaccines paired with additional protein antigens and adjuvants.

# Materials and methods

## Ethics statement

This study was carried out in accordance with the recommendations in the Guide for the Care and Use of Laboratory Animals of the National Institutes of Health. The protocol was approved by the Institutional Animal Care and Use Committee of Northwestern University (Protocol number: IS000011717).

## Poly (ethylene glycol)-block-poly (propylene sulfide) polymer synthesis

Poly (ethylene glycol)$_{17}$-block-poly-(propylene sulfide)$_{75}$ (PEG$_{17}$-*b*-PPS$_{75}$) copolymer was synthesized as previously reported (*Allen et al., 2018*; *Frey, 2021*). Briefly, monomethoxy PEG (molecular weight = 750 Da) was functionalized with mesyl chloride and substituted with thioacetate to produce a PEG acetate initiator for the polymerization of PPS.

## Bicontinuous nanosphere fabrication via flash nanoprecipitation

BCN was produced using the FNP technique as previously described (*Allen et al., 2018*). In brief, PEG$_{17}$-*b*-PPS$_{75}$ BCN polymer and any hydrophobic cargo were dissolved in THF and loaded onto a syringe. The hydrophilic cargo was loaded onto a separate syringe, and both impinged against one another through the confined impingement mixer into a scintillation vial. The vial was placed in a vacuum desiccator overnight to remove the organic solvent. NPs were filtered through the LH-20 column with PBS as an eluent and suspended in PBS for later use. A natural mixture of Mtb MA consists of α-, keto-, and methoxy forms of MA with chain lengths of 72–86 carbon atoms was obtained from Sigma (M4537, Sigma-Aldrich, St. Louis, MO) (*Van Rhijn et al., 2017*). MA encapsulation efficiency was analyzed using coumarin derivatization as previously described (*Shang et al., 2018*). MA was loaded at a concentration of 100 µg/mL of total BCN formulation and DiD was loaded at 5 µg/mL of total BCN formulation. MA loading capacity in BCNs was 2%. To measure the encapsulation efficiency of Ag85B, Ag85B-MA-BCN was centrifuged at 10,000 g for 10 min. The supernatant was collected and concentrated using Amicon Ultra-4 10 K centrifugal filter devices. To determine the unencapsulated protein concentration, the supernatant was incubated with Pierce BCA Protein assay reagent following the manufacturer protocol, and the absorbance was measured at 660 nm using a SpectraMax M3 multi-mode microplate reader (Molecular Devices, LLC). The percentage encapsulation efficiency was calculated as a percent of encapsulated protein to the total amount of protein added. Ag85B was encapsulated at a concentration of 140 µg/mL and loading capacity was found to be 2.88%.

## Poly lactic-co-glycolic acid nanoparticle (NP) fabrication

PLGA-NP was prepared by the Oil-in-Water single emulsion method. Briefly, the PLGA (20 mg in 800 µL dichloromethane) organic phase was emulsified using an ultrasonic processor with an aqueous phase containing 2 mL of polyvinyl alcohol (PVA) solution (2.5% w/v) to form an emulsion. The emulsion was then added dropwise into 4 mL of stirring 0.5% w/v PVA solution at room temperature to evaporate the organic solvent. NPs were collected after 4 hr of stirring followed by centrifugation at 10,000 x g for 10 min. After centrifugation, NPs were washed twice with cold water to remove

residual PVA and redispersed in PBS. Hydrophilic (Texas red-Dextran) or hydrophobic (MA) loading was performed by adding the payloads to the aqueous and organic phases, respectively.

## Nanocarrier characterization

The nanocarrier size (z-average diameter) and polydispersity were measured using dynamic light scattering (DLS) using a Nano 300 ZS Zetasizer (Malvern Panalytical, Malvern, U.K.). BCN and PLGA morphology was visualized with cryogenic transmission electron microscopy (cryo-TEM) as previously detailed (*Shang et al., 2018*). Transmission electron microscopy (TEM) was performed to image BCN using uranyl acetate as a negative stain as previously reported (*Bobbala et al., 2020*). BCN aggregate structure and internal morphology were characterized with SAXS. These studies were performed at the DuPont-Northwestern-Dow Collaborative Access Team beamline at Argonne National Laboratory's Advanced Photon Source with 10 keV (wavelength $\lambda$ =1.24 Å) collimated X-rays, as described previously (*Allen et al., 2018*).

## Confocal imaging

Confocal images of RAW 264.7 macrophages stained with lysosomal dye Lysotracker (green) and NucBlue stain (blue) following incubation with Texas Red-labeled PLGA-NP and BCN (red) for 8 hr. The cells were then imaged within a humidified chamber using a 63 X oil-immersion objective on an SP5 Leica Confocal Microscope using HyD detectors and lasers. Data analysis was performed using ImageJ software.

## Mouse strains

Human CD1 transgenic mice (hCD1Tg) in C57BL/6 background or MHC II-deficient background (*Felio et al., 2009*) and CD1b-restricted MA-specific TCR transgenic mice on $Rag^{-/-}$ background (DN1Tg/ hCD1Tg/$Rag^{-/-}$) (*Zhao et al., 2015*) were generated in our lab as previously described. P25-specific TCR transgenic mice (Jackson lab, *strain #:011005*) were crossed onto $Rag^{-/-}$ background. Both males and females were used.

## Antibodies and flow cytometry

For cell surface staining, cells were pre-incubated with 2.4G2 Fcγ RII/RIII blocking mAb for 15 min and then stained with the appropriate combinations of mAbs diluted in HBSS + 2% FBS for 30 min at 4 °C. Cells were analyzed on a BD FACS CantoII, or Cytek Aurora Spectral Cytometer, and data were processed with FlowJo software (TreeStar). The complete list of antibodies use is as follows: anti-human Vβ5.1 (LC4, Invitrogen), anti-TCRβ (H57-597, BioLegend), anti-CD44 (IM7, BioLegend), anti-Vβ11 (KT11, BioLegend), anti-CD45.2 (104, BioLegend), anti-CD45.1 (A20, BioLegend), anti-SiglecF (E50-2440, BD Bioscience), anti-CD45 (30-F11, BioLegend), anti-CD11c (N418, BioLegend), anti-CD11b (M1/70, BioLegend), anti-Ly6G (1A8, BioLegend), anti-CD19 (6D5, BioLegend), anti-NK1.1 (PK136, BioLegend), anti-CD3 (17A2, BioLegend), Zombie Red (BioLegend), anti-mouse CD185/ CXCR5 (L138D7, Biolegend), anti-mouse CD279/PD-1 (29 F.1A12, Biolegend), and anti-Ki67 (SolA15, Invitrogen).

## In vitro nanoparticle and lipid titration with MA-specific DN1 T cells

Nanoparticle and lipid titrations were performed as previously described (*Shang et al., 2018*). In brief, hCD1Tg MHC II$^{-/-}$ BMDCs were pulsed with varying quantities of nanoparticles of sonicated MA and then co-cultured for 48 hr with DN1 T cells isolated from lymph nodes (axial, brachial, submandibular, inguinal) of DN1Tg/hCD1Tg/$Rag^{-/-}$ mice. The supernatant was used for IFN-γ enzyme-linked immunosorbent assay (ELISA) performed as described previously (*Shang et al., 2018*) and cells were stained for activation.

## In vitro nanoparticle and lipid titration with human cells via ELISPOT assay

Hydrophobic polyvinylidene fluoride 96-well plates (Millipore, Bedford, MA) were coated with anti-human IFN-γ mAb 1-D1K (Mabtech, Cincinnati, OH) diluted 1:400 in PBS (Gibco, Waltham, MA) and incubated at 4 °C overnight. The following day, mAb was removed and each plate was washed and incubated at room temperature for 2 hr in the presence of media. Human monocyte-derived dendritic

cells and MA-specific M11 T cells were seeded 50,000 cells and 2000 cells per well, respectively, together with serially diluted MA and NPs. After 16 hr incubation, plates were washed and incubated with biotinylated anti-human IFN-γ (7-B6-1), diluted 1:3000 in PBS + 0.5% FBS, for 4 hr. Plates were then washed with PBS 5 times and incubated for 2 hr with ExtrAvidin-Alkaline phosphatase diluted 1:1000 in PBS + 0.5% FBS. Spots were visualized after incubation with the BCIP/NBT substrate for up to 20 minand then imaged with ImmunoSpot reader v.2.0.

### Mouse vaccination

For both IN and IT vaccination, mice were anesthetized using inhaled isoflurane. For IN vaccination, 25 μL of liquid was administered at the nostrils of the mice for a total of 1 μg of MA per mouse. For IT vaccination, mice were administered 25 or 50 μL of liquid for a total of 2.5 μg of MA with or without 2.0 μg of Ag85B. For BMDC vaccinations, hCD1Tg BMDCs were cultured in GM-CSF and IL-4 for 5 days (*Li et al., 2011*) and pulsed with MA-BCN (5 μg/mL of MA) or free MA (10 μg/mL of MA) for 24 hr. Cells were washed and resuspended in PBS and IT vaccinated at a dose of $1 \times 10^6$ BMDCs/mouse. Intravenous vaccination was administered by tail vein injection. Subcutaneous vaccination was administered between the shoulders over the neck portion of the mouse.

### Vaccination with attenuated Mtb

Attenuated Mtb strain (H37Rv Δ*panCD* Δl*euCD* referred to as mc² 6206) was kindly provided by Jacobs' Laboratory at Albert Einstein College of Medicine. Bacteria were grown as previously described in supplemented Middlebrook 7H9 media (*Sampson et al., 2004*; *Larsen et al., 2009*). Mice were vaccinated subcutaneously with $1 \times 10^6$ CFU in 100 μL of PBS. Mycobacterial burden quantification was performed by plating serial dilutions of lung or spleen homogenate on Middlebrook 7H11 agar plates (*Sampson et al., 2004*; *Larsen et al., 2009*).

### T cell Adoptive transfer and cell preparation

DN1 or p25 T cells were isolated from lymph nodes of DN1Tg/hCD1Tg/Rag$^{-/-}$ or P25Tg/Rag$^{-/-}$ mice, respectively. Cells were labeled with CellTrace Violet reagent (Invitrogen) as per manufacturer's instructions and $3–5 \times 10^6$ cells were injected into hCD1Tg mice either 1 day before vaccination (short term) or 6 weeks post-vaccination (long term). On day 7 post-adoptive transfer, draining lymph nodes (axial, brachial, mediastinal), lung, and spleen were obtained. Single-cell suspensions were prepared by mechanical disruption in HBSS/2% FBS. The lung was digested with collagenase IV (1 mg/ml; Sigma-Aldrich) and DNase I (30 μg/ml; Sigma-Aldrich) for 30 min at 37 °C before the disruption.

### Biodistribution imaging

BCN was loaded with DiD dye (Thermo Fisher) and administered to mice IT at 4 hr, 24 hr, 48 hr, 6 days, and 6 weeks prior to the final time point, at which all mice were sacrificed and analyzed together. Lung, spleen, and draining lymph nodes were harvested and imaged using a near-IR in vivo Imaging System (IVIS; Center for Advanced Molecular Imaging, Northwestern University). The single-cell suspension was then prepared from indicated organs and cells were stained for flow cytometric analysis.

### Alveolar macrophage enrichment and T cell co-culture

hCD1Tg mice were immunized IT with either BCN or MA-BCN (2.5 μg of MA/mouse). 6 weeks later, lungs were isolated and single-cell suspension was stained with anti-SiglecF-PE (E50-2440, BD Bioscience). SiglecF-positive cells were enriched using an anti-PE MultiSort Kit (Miltenyi Biotec). $1 \times 10^5$ flow through and AM-enriched fractions were co-cultured with $1 \times 10^5$ hCD1Tg BMDCs and $3 \times 10^5$ DN1 T cells for 48 hr and cells were stained for the expression of activation markers.

### Generation and vaccination of BM chimeras

Bone marrow cells from DN1Tg/hCD1Tg/Rag$^{-/-}$ (CD45.2) and hCD1Tg mice (CD45.1) were depleted of mature T cells using anti-Thy-1.2 (AT83.A-6) plus rabbit complement (Cedarlane Laboratories). Equal numbers of cells ($5 \times 10^6$) were adoptively transferred into hCD1Tg (CD45.1) mice irradiated with 900 rads. After 6 weeks, BM chimeric mice were intratracheally vaccinated with either 2.5 μg of MA in MA-BCN or an equivalent volume of PBS, and the activation status of DN1 T cells were monitored in the blood 1–2 times per week until experiment completion.

## RNA-seq analysis

CD44$^+$CD62L$^+$ and CD44$^-$CD62L$^+$ DN1 T cells were sorted from LNs of DN1-hCD1Tg BM chimera mice vaccinated with MA-BCN at 6 weeks post-vaccination by BD FACS Aria with purity >98%. RNA was extracted with RNAeasy mini kit (Qiagen). Libraries were generated by using the Illumina Truseq preparation kit and sequenced on HiSeq4000. Reads were analyzed with the Ceto pipeline (*Bartom et al., 2022*) using STAR and HTseq for alignment on the mm10 mouse genome and reading counting, as described previously (*Weng et al., 2021*). Paired differential expression analysis was performed using DESeq2 to account for original mouse sourcing (*Love et al., 2014*). All downstream analysis was performed in R and figures were generated using ggplot. Gene enrichment analysis was performed using Metascape (*Zhou et al., 2019*). Comparisons to other T cell profiles were performed using raw transcriptome data from the ImmGen database (*Yoshida et al., 2019*), and two additional publications for follicular helper (T$_{FH}$) (*Lahmann et al., 2019*), exhausted (Texh) (*Man et al., 2017*). Analysis was performed as described above to obtain gene expression fold change values relative to internal naïve conditions.

## Statistical analysis

Statistical analyses were performed using Prism software 9 (GraphPad Software, Inc). Multi-group comparisons were done using two-way analysis of variance (ANOVA) with significance determined by Tukey's multiple comparisons. Where appropriate, outliers were identified through Grubbs method with alpha = 0.05. Statistically significance denoted as ns = not significant, *p<0.05, **p<0.01, ***p<0.001, and ****p<0.0001.

## Acknowledgements

We thank Ying He for her support in maintaining laboratory reagents and equipment, Elizabeth Bartom for the use of her RNA-seq analysis pipeline, and Michael Kleyman for his advice on comparing gene expression among multiple T cell populations. We would like to thank the Center for Advanced Molecular Imaging (CAMI) for their help in performing IVIS and The Northwestern University's Atomic and Nanoscale Characterization Experimental Center (NUANCE) Keck-II facility for their help with NP characterization. Ag85B from *Mycobacterium tuberculosis* H37Rv was obtained through BEI Resources, NIAID, and NIH. SAXS experiments were performed at the DuPont-Northwestern-Dow Collaborative Access Team (DND-CAT) located at Sector 5 of the Advanced Photon Source. DND-CAT is supported by Northwestern University, The Dow Chemical Company, and DuPont de Nemours, Inc This research used resources of the Advanced Photon Source; a U.S. Department of Energy (DOE) Office of Science User Facility operated for the DOE Office of Science by Argonne National Laboratory under Contract No. DE-AC02-06CH11357. Data was collected using an instrument funded by the National Science Foundation under Award No. 0960140. Funding: This work was supported by the following grants: National Institutes of Health grant R01AI057460 (CRW), R01AI145345 (CRW, EAS), National Institutes of Health grant F30AI157314 (EM), National Institutes of Health grant T32GM008152 (EM), and National Institutes of Health grant R01AI146072 (CS).

## Additional information

### Competing interests

Chyung-Ru Wang: Reviewing Editor *eLife*. The other authors declare that no competing interests exist.

### Funding

| Funder | Grant reference number | Author |
|---|---|---|
| National Institutes of Health | R01AI057460 | Chyung-Ru Wang |
| National Institutes of Health | R01AI145345 | Evan A Scott<br>Chyung-Ru Wang |

| Funder | Grant reference number | Author |
|---|---|---|
| National Institutes of Health | F30AI157314 | Eva Morgun |
| National Institutes of Health | R01AI146072 | Chetan Seshadri |
| National Institutes of Health | T32GM008152 | Eva Morgun |

The funders had no role in study design, data collection and interpretation, or the decision to submit the work for publication.

## Author contributions

Eva Morgun, Investigation, Methodology, Writing – original draft, Writing – review and editing; Jennifer Zhu, Sharan Bobbala, Melissa S Aguilar, Junzhong Wang, Investigation, Methodology, Writing – review and editing; Sultan Almunif, Yongyong Cui, Investigation, Methodology; Kathleen Conner, Liang Cao, Investigation; Chetan Seshadri, Supervision, Writing – review and editing; Evan A Scott, Chyung-Ru Wang, Conceptualization, Supervision, Funding acquisition, Writing – review and editing

## Author ORCIDs

Eva Morgun ⓘ https://orcid.org/0000-0003-2386-4007
Chetan Seshadri ⓘ http://orcid.org/0000-0003-2783-7540
Chyung-Ru Wang ⓘ https://orcid.org/0000-0002-9835-4784

## Ethics

This study was carried out in accordance with the recommendations in the Guide for the Care and Use of Laboratory Animals of the National Institutes of Health. The protocol was approved by theInstitutional Animal Care and Use Committee of Northwestern University (Protocol number: IS000011717).

Reviewer #1 (Public Review): https://doi.org/10.7554/eLife.87431.3.sa1
Reviewer #2 (Public Review): https://doi.org/10.7554/eLife.87431.3.sa2
Author Response https://doi.org/10.7554/eLife.87431.3.sa3

# Additional files

## Supplementary files
• MDAR checklist

## Data availability

All data are available in the main text or the supplementary materials. RNA-seq data described in this publication have been deposited in NCBI's Gene Expression Omnibus and are accessible through GEO Series accession number GSE226075. Generated transgenic mice and nanoparticles are available upon request upon signing of the Material Transfer Agreement (MTA).

The following dataset was generated:

| Author(s) | Year | Dataset title | Dataset URL | Database and Identifier |
|---|---|---|---|---|
| Morgun E, Zhu J, Almunif S, Bobbala S, Aguilar MS, Wang J, Cui Y, Cao L, Seshadri C, Scott EA, Wang C | 2023 | Mycolic acid nanoparticle vaccination leads to lipid antigen persistence and unique differentiation of mycobacterial-specific T cells | https://www.ncbi.nlm.nih.gov/geo/query/acc.cgi?acc=GSE226075 | NCBI Gene Expression Omnibus, GSE226075 |

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
