## [Editor Report · eLife assessment]

The authors generate a new formulation built upon a previous nanoparticle platform to generate a new system termed bicontinuous nanospheres (BCN), allowing for the dual incorporation of lipid and protein antigens. The authors generate mycolic acid (MA)-loaded BCN and perform a series of characterization studies to demonstrate the superior performance of this new formulation relative to the original one in terms of antigen persistence, a quality needed to sustain responses after vaccination. This work provides **important** new insights relevant to the TB vaccine field and it suggests that alternative antigens to proteins could be used in TB vaccine formulations. The data are **convincing** and will be of interest to individuals working on tuberculosis, vaccines and basic immunology.

---

## [Referee Report · Reviewer #1 (Public Review)]

It is well established that tuberculosis (TB), which is caused by Mycobacterium tuberculosis (Mtb), is a leading cause of mortality and morbidity worldwide. However, the only vaccine licensed against tuberculosis is Bacille Calmette Guerin (BCG), has been around for nearly a century, and has limited efficacy in adults. Herein, the authors sought to investigate the effectiveness of a nanoparticle-based formulation of a subunit vaccine composed of Mtb lipid and protein antigens. The authors found that they were able to load the lipid, mycolic acid, into their nanoparticles without disrupting the architecture and that the loaded particles activated T cells both in vitro and in vivo. Moreover, when they vaccinated with particles loaded with both lipid and protein antigens, they found that the lipid antigen persisted, and mycolic acid-specific T cells were able to be activated 6 weeks post-vaccination, in contrast to peptide-specific T cells. The authors investigated further and found that persistence required the nanoparticle encapsulation, rather than free lipid, and that it was independent of route (intratracheal, intravenous, or subcutaneous) of administration. To address the mechanisms underlying antigen persistence, the authors loaded the nanoparticles with a dye and demonstrated that the nanoparticle encapsulated lipid antigen was primarily stored in lung alveolar macrophages and that CD1b+ dendritic cells presented the antigen to mycolic acid specific T cells. Finally, the authors conducted mixed bone marrow chimera studies to examine the phenotype of the mycolic acid specific T cells and found that the memory T cell population phenotypically resembled T follicular helper, regulatory T cells, and exhausted T cells. Interestingly, while a large percentage of these lipid antigen specific T cells in the lymph nodes, lung and spleen were CXCR5+PD1+, the cells were still proliferating (Ki67+). Overall, this is a comprehensive study that has the potential to significantly enhance the field.

---

## [Referee Report · Reviewer #2 (Public Review)]

The work presented here by Morgun et al is performed in the context of vaccine development, a field especially active in the context of tuberculosis (TB). The generation of a new vaccine either enhancing or replacing the 100-year-old BCG is urgently needed.

Most subunit vaccines integrate protein antigens formulated with adjuvants and there are few examples of the performance of subunit vaccines integrating lipid antigens. Considering the hydrophobic and lipid nature of the mycobacterial cell envelope studies, assessing the suitability of mycobacterial lipids in vaccine formulations may contribute to generate new vaccines to tackle the disease.

The mycobacterial lipid antigens under study are mycolic acids (MA), which are located at the cell wall covalently linked to arabinogalactan. These lipids carry extremely long chain fatty acids of up to 60-90 carbons.

The group has previously shown that formulating MA into micellar nanocarriers and vaccinating mice intranasally it could activate CD1-restricted T cells. However, this formulation did not allow for the incorporation of protein antigens.

This work is novel, and it brings new data of high relevance for the TB vaccine field pointing to alternative formulations and antigens and immune mechanisms.

Authors assay different routes of vaccination but the main results are obtained using non-conventional vaccination routes. Although, it maybe out of the scope of the paper, no protection studies are provided.

---

## [Author Response]

The following is the authors’ response to the original reviews.

We would like to thank both reviewers and the editor for their time and effort in carefully reviewing and comprehending our manuscript. We are grateful for their thorough assessment, as well as the insightful questions and suggestions they have provided. We have taken into account the questions and comments raised by the reviewers, and we have incorporated the necessary revisions accordingly. In the following pages the reviewers’ comments are italicized. Our replies are in normal script.

In addition to revisions suggested by reviewers we also added a new summary schematic (Fig 8) and minor changes to acknowledgments.

**Reviewer 1**
This is a very strong study with few concerns. Regarding DN1+ T cell function, the authors assessed IFN-γ and activation markers, but it is unclear if the cells are polyfunctional (produced high levels of other cytokines at 6 weeks) or if there were changes in the humoral response (serum Ab titers or size/ number of germinal centers.)

Thank you for your thorough assessment of our work and your kind comments.

a. We observed a decreased IFN-γ and TNF-α production in antigen experienced DN1 T cells compared to naïve DN1 T cells, which is consistent with findings in Tfh cells.

b. We tested for anti-MA IgM and IgG production but did not observe an increase in these antibodies in the vaccinated setting. It is possible that additional inflammatory stimulation, such as from an adjuvant or infection, may be necessary to trigger sufficient antibody level for detection using ELISA.

c. We did not measure the number or size of germinal centers in this study, but future investigations could explore this aspect.

**Reviewer 2**
1. Authors elaborate the introduction solely highlighting the relevance of antigen persistence in the context of vaccination. However, it is well known that several mycobacterial antigens (Lipids and proteins) can cause detrimental responses when overexposed to the immune system. In this regard, it would be appropriate to introduce the possibility of the occurrence of exhaustion when prolonged exposure to antigens is happening, which is the main theme of this paper.

Thank you for bringing these points to our attention. We have added a paragraph in the discussion section (page 15-16, line 372-386), addressing the implications of our findings in relation to exhaustion in the context of antigen persistence during chronic viral infections. We have also provided an example involving the lipid trehalose 6,6’-dibehenateled (TDM), a known virulence factor for Mtb, which has been utilized in several subunit vaccines without demonstrating significant toxicity.

2. Authors need to provide more information about the source of MA. It is briefly mentioned in the materials and methods section that it was obtained from Sigma. If that is the case, it would be ideal to show the integrity of the polysaccharide in term of balance and abundance between different MA species.

We obtained M. tuberculosis MA from Sigma, which comprises α-, keto-, and methoxy MA forms with an average combined lipid tail length of 80 carbons. MA-specific T cells preferentially recognize these three forms of MA have been identified in humans. We have provided more detailed information regarding the MA in the Materials and Methods section (page 17, line 429-431).

3. Building up on the previous comment, MA is a complex mixture of polysaccharides including multiple lengths of fatty acids and modifications. Could the authors comments on the potential variability of MA structure and potential impact on immune responses?

The binding capacities of Group 1 CD1-restricted T cells can be influenced by various factors, including specific head groups, lipid tail length, and structure of the lipid tail. Notably, DN1 T cells have been shown to have higher binding affinities towards keto and methoxy MA, while displaying weaker binding to α-MA (Van Rhijn et al., 2017, Eur. J. Immunol. 47:1525). In our study, we successfully utilized a mixture of MA to activate DN1 T cells, indicating that the required subtypes of MA were present in sufficient quantities to elicit this activation. In future investigations focusing on the polyclonal immune response, incorporating a mixture of MA and possibly other Mtb lipid antigens will enable a broader spectrum of T cell activation. This, in turn, is expected to enhance the overall effectiveness and robustness of protection in challenge experiments.

4. How do the authors explain the lack of stimulation of cell proliferation induced by MA-PLGA formulation? Does this result contradict previous findings?

This study represents the first instance of utilizing PLGA as a delivery system for a lipid antigen via a pulmonary vaccine route, despite its previous applications in numerous other vaccine formulations. Therefore, we do not think our findings contradicts any existing research in the field. It is worth noting that the immunogenicity of PLGA can be influenced by the specific polymer chemistry and formulation, which may account for potential variations in the observed effects. We have added additional text to the discussion (page 13, line 310 – 313) to address this point.

5. Fig 3. Authors switch to IT administration simply arguing against the limitation of IN delivery regarding its low volume. However, administration via IN could be done in an iterative manner. According to this change, this reviewer asks whether the performance of MA-PLGA could now be comparable to BCN-MA using IT instead.

PLGA possesses an inherent background adjuvant effect, which may not be ideal for precisely stimulating group 1 CD1-restricted T cells, as a considerable proportion of these T cells exhibit some level of autoreactivity (Li, et al, 2011, Blood 118:3870, De Lalla et al., 2011, Eur. J. Immunol. 41:602; de Jong et al, 2010, Nat. Immunol. 11:1102). Notably, our observations revealed that blank PLGA-NP exerted a significant stimulatory effect on both mouse (DN1) and human (M11) MA-specific T cells (Fig. 2A-D). This underscores the advantage of the BCN system, which lacks detectable adjuvant effects and enables a more controlled, dose-dependent augmentation of T cell responses with increasing concentrations of loaded MA. Therefore, we did not further evaluate the impact of PLGA-MA using the IT route of vaccination.

6. What would be the reasons of the no role of encapsulating NP in the persistence of MA?

In this study, we have provided evidence to support the notion that encapsulation plays a role in antigen persistence, as demonstrated in Fig. 5A-C. Specifically, we directly compared the persistence of MA when delivered encapsulated in BCNs versus without encapsulation in BCNs, using DC pulsing and IT vaccination as the delivery methods. Our results indicate that at 6 weeks post-vaccination, MA encapsulated in BCNs can activate DN1 T cells, while free MA does not. These findings may initially appear to be contradictory to those depicted in Fig. 5D-F, where antigen persistence is observed following vaccination with attenuated Mtb. However, we propose that the attenuated Mtb bacteria may function similarly to nanoparticles by encapsulating and containing MA, thereby facilitating its persistence within the host. We appreciate the opportunity to clarify these points (page 15, line 364-367). Encapsulation within PEG-PPS NP may also contribute to two additional mechanisms. First, we have demonstrated that PEG-PPS NPs target myeloid cell populations (Burke et al., 2022, Nat. Nano. 17:319), such as alveolar macrophages, that can serve as antigen persistence depots as well as present CD1b/MA complexes on their surfaces. NPs allow more efficient delivery to these cells, whereas otherwise the lipid would bind to albumin, HDL, LDL, and other lipid carriers in blood for a broader, non-specific biodistribution, which would include cells less efficient at antigen persistence or presentation. Second, we previously demonstrated that the BCN nanostructure is highly stable within cells, supporting a slow intracellular release (Bobbala et al., 2020, Nanoscale 12:5332). This could assist with a more sustained presentation of lipid antigen by targeted cells in contrast to free form lipid or NPs (like PLGA) that rapidly degrade within cells. Indeed, low levels of fluorescently tagged BCNs were still detectable 6 weeks post-vaccination (Fig. 6B). Our future studies will further investigate this hypothesis.

7. Authors need to discuss to what extent the MA location into AM is route dependent.

The localization of MA within alveolar macrophages (AMs) in the lung is likely specific to intratracheal (IT) vaccination. Therefore, mice vaccinated subcutaneously (SC) or intravenously (IV) may possess distinct antigen persistence depots. We have made modifications to the discussion section to further emphasize this point (page 15, line 359-364).

8. Also, AM are programmed to sustain low immune responses because of their unique location in the lung. In fact, Mtb uses this to replicate while immune response is mounted. In this regard, accumulation of MA into this compartment may not be relevant for the overall immune response. In other words, what would be the contribution of this population to the T cell activation?

It is likely that AMs primarily function as antigen depots and do not directly contribute to the activation of DN1 T cells. This assertion is supported by our findings, as co-culturing AMs with DN1 T cells alone did not result in T cell activation (Fig. 6E). However, we observed that the presence of hCD1Tg-expressing bone marrow-derived dendritic cells was necessary for DN1 T cell activation in vitro, which likely reflects a similar phenomenon occurring in vivo.

9. Could the T cells responses measured be due to the reduced fraction of DC loaded with BCN-MA at initial time points?

Regarding the T cell response observed in Fig. 5A-C, where we used DCs to deliver either free MA or MA-BCN, we took steps to address potential differences in loading capacity between the two at initial time points. Specifically, DCs were pulsed with a concentration of 10 μg/mL for free MA and 5 μg/mL of MA-BCN (the figure legend has been modified to clarify this point, page 37, line 962 - 963). To ensure approximate equivalence in loading, we examined the immune response one week after vaccination and found no statistically significant difference between the two methods.